

# Effects of black carbon and boundary layer interaction on surface ozone in Nanjing, China

Jinhui Gao[1,2,3,4], Bin Zhu[1,2,3,4], Hui Xiao[5], Hanqing Kang[1,2,3,4], Chen Pan[1,2,3,4]

[1]Key Laboratory for Aerosol-Cloud-Precipitation of China Meteorological Administration, Nanjing University of Information Science & Technology, Nanjing, China
[2]Collaborative Innovation Centre on Forecast and Evaluation of Meteorological Disasters, Nanjing University of Information Science & Technology, Nanjing, China
[3]Key Laboratory of Meteorological Disaster, Ministry of Education (KLME), Nanjing University of Information Science & Technology, Nanjing, China
[4]Joint International Research Laboratory of Climate and Environment Change (ILCEC), Nanjing University of Information Science & Technology, Nanjing, China
[5]Institute of Tropical and marine Meteorology, China Meteorological Administration, Guangzhou, China

*Correspondence to*: Bin Zhu (binzhu@nuist.edu.cn)

**Abstract.** As an important solar-radiation absorbing aerosol, the effect of black carbon (BC) on surface ozone, by influencing photolysis rate, has been widely discussed by "offline" model studies. However, BC-boundary layer (BL) interactions also influence surface ozone. Using the "online" model simulations and processes analysis, we demonstrate the significant impact of BC-BL interaction on surface ozone. The absorbing effect of BC heats the air above the BL and suppresses BL development, which eventually leads to changes in the contributions of ozone through chemical and physical processes (photochemistry, vertical mixing, and advection). Different from previous "offline" model studies, BL suppression leads large amounts of ozone precursors being confined below the BL which offsetting the influence from the reduction of photolysis rate, thus enhancing ozone photochemical formation before noon. Furthermore, the changes in physical process show a more significant influence on surface ozone. The weakened turbulence entrains much less ozone from the overlying ozone-rich air down to surface. As a result, the net contribution of ozone from physical and chemical processes leads to surface ozone reduction before noon. The maximum reduction reaches to 16.4 ppb at 12:00. In the afternoon, the changes in chemical process are small which influence inconspicuously to surface ozone. However, physical process still influences the surface ozone significantly. Due to the delayed development of the BL, less vertically mixed BL continues to show an obvious ozone gradient near the top of the BL. Therefore, more ozone aloft can be entrained down to the surface, offsetting the surface ozone reduction. Comparing all the changes in the contributions of processes, the change in the contribution of vertical mixing plays a more important role in impacting surface ozone. Our results show the great impacts of BC-BL interactions on surface ozone. And more attention should be paid on the mechanism of aerosol-BL interactions when we deal with the ozone pollution control in China.



# 1 Introduction

Black carbon (BC) aerosol, also known as soot, is primarily formed by incomplete combustion of carbonaceous fuels, diesel fuels, and biomass (Bond et al., 2004). BC accounts for a small fraction, less than 15%, of the total mass concentration of aerosol particles in atmosphere over urban areas (Yang et al., 2011). However, it is of great interest because of its significant

influences on global radiation balance (Chameides and Bergin, 2002), both directly, by absorbing solar radiation (Liao and Seinfeld, 2005; Ramanathan and Carmichael, 2008), and indirectly, by affecting cloud formation (Lohmann and Feichter, 1997; Fan et al., 2015). Owing to such impacts on radiation, BC plays an important role in global and regional climate change (Jacobson, 2001; Bond et al., 2013), weather (Qian et al, 2003; Saide et al., 2015), and the atmospheric environment (Li et al., 2005; Ding et al., 2016; Peng et al., 2016). In addition, the BC aging mechanism (Qiu et al., 2012), the

performance of BC in heterogeneous reactions (Lei et al., 2004; Li et al., 2012), and its impact on human health (Atkinson et al., 2012) have been the focus of significant research in recent years.

Tropospheric ozone is a typical secondary air pollutant (Crutzen, 1973). It has important environmental effects on the atmosphere (Monks et al., 2015) especially in the boundary layer (BL). The impact of aerosols, especially BC, on surface ozone has been attracting much attention from researchers. Dickerson et al. (1997) reported that BC decreases surface ozone

concentration by reducing photolysis rates. Jacobson (1998) suggested that aerosols containing BC cores reduced photolysis rates and resulted in a decrease of ozone concentration by 5%−8% at ground level in Los Angeles. Castro et al. (2001) found a strong reduction in photolysis rate (10%−30%) due to BC-containing aerosols. They also reported that this photolysis rate reduction led to a decrease of surface ozone in Mexico City. Similar results have been found in other studies simulating the effects of BC on surface ozone in other places around the world (Li et al., 2005; Li et al., 2011).

In addition to reducing photolysis rates, the global warming effect of BC is significant, preceded only by $CO_2$ (Jacobson, 2002). Incident solar radiation is absorbed by BC in the atmosphere, leading to the air aloft being heated and the temperature being raised. Conversely, air at low levels is cooled and the temperature is decreased. Under this condition, atmospheric stability is increased and the development of BL is suppressed during the daytime. Using an atmospheric model, Ding et al. (2016) demonstrated this effect and suggested that such BC-BL interactions will enhance the occurrences of haze pollution

episodes. Owing to the close relationship between ozone and BL development during the daytime (Zhang and Rao, 1999; Zhu et al., 2015), BC-BL interactions may also influence surface ozone; however, relevant studies on this phenomenon are still lacking.

As one of the most developed regions in China, the Yangtze River Delta (YRD) has reported severe haze (Ding et al., 2013a) and ozone pollution (Tie et al., 2013) in recent decades. Nanjing is the capital of Jiangsu province, which is an important

industrial and economic centre in the YRD region. Previous studies have reported that the BC (Zhuang et al., 2014) and ozone (Ding et al., 2013b) concentrations are relatively high in Nanjing in October. In this study, measured data of BC and ozone, which were obtained in Nanjing in October 2015, were used to show variations in surface ozone under the conditions of high BC concentration and low BC concentration during daytime. Furthermore, the fully coupled "online" model system



Weather Research and Forecasting with Chemistry (WRF-Chem) was applied to simulate the air pollutants (ozone, $PM_{2.5}$, and BC) in YRD in October 2015. With the consideration of the aerosol-BL feedback mechanism in the model system, we demonstrate the mechanism of the BC-BL interaction affecting surface ozone in Nanjing.

## 2 Model setting and data description

The Weather Research and Forecasting (WRF) model coupled with Chemistry (WRF-Chem), which is widely used to evaluate the impacts of aerosols on radiation (Zhang et al., 2010; Forkel et al, 2012), is a fully coupled online 3D Eulerian chemical transport model considering both chemical and physical processes. We used version 3.4 in this study, and detailed descriptions of the meteorological and chemical aspects of WRF-Chem can be found in Skamarock et al. (2008) and Grell et al. (2005). Regarding simulation settings, two nested domains (Figure 1) were set up with horizontal resolutions of 36 km

and 12 km, and grids of $99 \times 99$ and $99 \times 99$ for the parent domain (D1) and nested domain (D2), respectively. The parent domain (D1), centred at (119.0 °E, 31.5 °N), covered most of China and the surrounding countries and ocean. The corresponding simulations provided meteorological and chemical boundary conditions for the nested domain (D2), which covered most of Eastern China.

There were 38 vertical layers from the surface up to a pressure limit of 50 hPa, of which 12 levels were located below the

lowest 2 km to fully describe the vertical structure of the BL. Carbon-Bond Mechanism Z (CBM-Z), which includes 133 chemical reactions for 53 species and extends the framework to function for a longer time and at larger spatial scales than its predecessor; Carbon-Bond IV, was used as the gas-phase chemical mechanism (Zaveri and Peters, 1999). The corresponding aerosol chemical mechanism was the Model for Simulating Aerosol Interactions and Chemistry (MOSAIC) with 8 bins (Zaveri et al., 2008), which is extremely efficient and does not compromise accuracy in aerosol model calculations. Other

major model configuration options are listed in Table 1.

The National Centres for Environment Prediction (NCEP) final (FNL) Operational Global Analysis data files were used to provide the initial and boundary meteorological conditions for our simulations. The initial and boundary chemistry conditions were provided by the output of the Model of Ozone and Related Chemical Tracers (MOZART-4; Emmons et al., 2010). Both anthropogenic and natural emissions were inputted into the model system. Anthropogenic emissions were

derived from the Multi-resolution Emission Inventory for China (MEIC) database (http://www.meicmodel.org/). MEIC contains both gaseous and aerosol species, including $SO_2$, $NO_X$, $NH_3$, CO, VOCs (volatile organic compounds), BC, OC (organic carbon), $PM_{10}$, and $PM_{2.5}$. Biogenic emissions were calculated using the Model of Emission of Gas and Aerosols from Nature (MEGAN; Guenther et al., 2006).

Two parallel experiments were conducted to investigate our subject: (1) simulation with aerosol feedback considering both

the direct and indirect radiation effects from all chemical species (Exp_WF), in which aerosol optical properties were calculated at each time step and then coupled with the radiative transfer model for both short and long wave radiation (Iacono et al, 2008); (2) simulation with aerosol feedback, excluding BC (Exp_WFexBC), in which only the optical property



of BC was subtracted when calculating the shortwave radiation and optical properties of all other aerosols were retained and calculated in the same manner as in experiment EXP_WF (Wang et al., 2016). The two experiments started at 00:00 UTC on 1 October 2015 and ended at 00:00 UTC on 26 October 2015. In order to reduce the influence of initial conditions, the first 9 days were designated as the spin-up period.

We collected observational data on both meteorology (temperature, wind direction, and wind speed) and air pollutants (Ozone, $NO_2$ and $PM_{2.5}$) in five cities (Hefei, Maanshan, Nanjing, Zhejiang, and Wuxi, Figure 1) in October 2015 to evaluate the model performance. Temperature, wind direction, and wind speed, with a temporal resolution of 3 h, were obtained from Meteorological Information Comprehensive Analysis and Process System (MICAPS). And these data were measured by the national surface observation network operated by the China Meteorological Administration (CMA). Data on the hourly

concentrations of Ozone, $NO_2$, and $PM_{2.5}$ were downloaded from the publishing website of China National Environmental Monitoring Centre (http://113.108.142.147:20035/emcpublish). These air pollutants and three other pollutants ($SO_2$, $PM_{10}$, and CO) were measured by the national air quality monitoring network operated by the Ministry of Environmental Protection of China. The China National Environmental Monitoring Centre is responsible for ensuring data quality. More information of the air pollution measurements are available in Wang et al. (2014). In addition, measurements of shortwave

irradiance and BC concentrations in the northern suburb of Nanjing (near the ozone monitoring site) were taken. Shortwave irradiance was measured using a Pyranometer (MS-802F of EKO instrument, Japan). The measurement accuracy is 1 W m$^{-2}$ and the temporal resolution is 1 min. More information is available in the instrument manual (https://eko-eu.com/files/PyranometerManual20160926V11.pdf). Hourly concentrations of BC were measured by using an Aethalometer (model AE-33 of Magee Scientific, USA). The sampling time interval was set to 1 min and the inlet flow was set to 5 L min$^{-}$

$^1$. More information is available in Drinovec et al. (2015).

## 3 Result and Discussions

### 3.1 Model evaluation

To evaluate model performance, the measured and simulated variables from 10 to 26 October 2015 were compared (Figure 2). In addition, shortwave and BC concentration data observed in Nanjing were used to evaluate the corresponding model

predictions.

#### 3.1.1 Meteorology evaluation

Regarding meteorological factors, which highly impact transport, deposition, and transformation in the atmosphere (Li et al., 2008), a good performance of meteorological conditions will guarantee accuracy in the air pollutant simulations. In this study, the observed temperature, wind direction, and wind speed were used for evaluating the meteorological model. In

addition, the shortwave irradiance measured in Nanjing was also used to evaluate the simulation of shortwaves. Model performance statistics, including index of agreement (IOA), mean bias (MB), and the root mean square error (RMSE), are





shown in Table 2, along with benchmarks derived from Emery et al. (2001). The IOA of wind direction (WD) was based on the calculation suggested by Kwok et al. (2010), which considers the nature of WD, and others were calculated following the approach of Lu et al. (1997).

WD showed similar patterns between the simulation and observations, but MB values of WD in Zhenjiang, Hefei, and Wuxi

were beyond the benchmark. However, the high values of IOA showed good agreement, indicating that the simulation successfully captured the wind direction during the period. Wind speed (WS) showed a slight over-estimation in Hefei and Wuxi with positive MBs of 0.72 and 0.87. Furthermore, the WS statistics are acceptable, especially IOAs and RMSEs, which are within the scope of the benchmarks. The negative MBs of temperature at 2 m above the surface (T2) indicate that the model predictions slightly underestimated T2. High values of IOAs in these cities indicate acceptable agreements between

measurements and simulations. The short-wave radiation (SW) time series showed similar patterns between the simulation and measurements in Nanjing. The high IOAs and low biases (MB and RMSE) suggest satisfactory model performance on SW, which is very important for the determination of photochemistry and BC feedback mechanism.

### 3.1.2 Air pollutant evaluation

Measured hourly concentrations of $O_3$, $NO_2$, and $PM_{2.5}$ in the five cities were collected to evaluate the model performance on

air pollutants. In addition, hourly BC concentrations measured in Nanjing were also used for evaluating the BC simulation. Statistical metrics on air pollutants, which include IOA, mean normalized bias (MNB), and mean fractional bias (MFB), are shown in Table 3, along with benchmarks derived from EPA (2005, 2007) .

For ozone, MNBs in Nanjing and Wuxi were slightly beyond the benchmarks. However, the time series shows similar patterns between the simulation and measurements (Figure 2), reflected by the high values of IOA. As an important ozone

precursor, a good model performance on $NO_2$ is necessary. More than 0.7 of IOA values reflect the good agreement between measurements and simulations. The values of MNBs and MFBs are close to those of other studies in this region in October (Hu et al., 2016). Owing to its complex chemical composition, accurate simulation of $PM_{2.5}$ was difficult. The IOA values of $PM_{2.5}$ were lower than those of $O_3$. However, MFBs were within the benchmarks, indicating that the model predictions of $PM_{2.5}$ are acceptable. For the BC time series, similar patterns were found between the simulation and observation.

Considering that the MFB was within the benchmarks for $PM_{2.5}$ and the relevant high value of IOA (0.67), the model predictions of BC were deemed acceptable.

In general, based on the comparisons between measurements and simulations of both meteorology and air pollutants, the similar time series patterns and acceptable statistical results demonstrate the good simulation capability of the model system in capturing the meteorological and chemical features. In this case, the meteorological and chemical features in Eastern

China from 10 to 26 October 2015 were well reproduced, which is favourable for our analysis and discussions.



## 3.2 Observational and model results of BC and ozone

Analysing relevant observation data, ozone and BC concentrations on sunny days were selected. Subsequently, ozone concentrations were distributed into two sets (Figure 3a) based on the daytime mean BC concentrations being higher (black) and lower (red) than the monthly mean concentration (~3 μg m$^{-3}$). When BC concentration was higher, the increase in ozone

concentrations occurred later than when BC was lower during 10:00 to 14:00. At 12:00, the difference between the two patterns of ozone concentrations reached to the maximum value. Because of the limited measurements, the formation of changes in ozone and the relationship between the BC-BL interaction and surface ozone could not be validated. In this case, numerical simulations using the online-coupled chemistry transport model WRF-Chem provides an effective method to analyse and discuss this subject.

With the application of WRF-Chem, Ding et al. (2016) reported the suppression of BL development induced by the warming effects of BC. By analysing our model outputs, the maximum changes in  mean BL-Height  ($\Delta BLH\_MAX$) over Nanjing were calculated. $\Delta BLH\_MAX$ is defined as the maximum difference of hourly mean BLH over Nanjing between Exp_WFexBC and Exp_WF during morning and noon [Equation (1)].

$$\Delta BLH_{MAX} = \max(\overline{BLH}_{WFexBC}^{t=10:00} - \overline{BLH}_{WF}^{t=10:00}, \dots \dots, \overline{BLH}_{WFexBC}^{t=12:00} - \overline{BLH}_{WF}^{t=12:00}), \tag{1}$$

In Eq. (1), $\overline{BLH}$ is the mean boundary layer height over Nanjing. The simulated mean BC columns (from the surface to 2 km) over Nanjing, during the occurrence of $\Delta BLH\_MAX$, were also calculated. The relationship between the two variables is shown in Figure 3b. Similar to Ding's study, the positive values suggest that BL development is suppressed by the warming effect of BC. This effect is more significant at higher BC concentrations. For example, the BC column was more than 9 mg m$^{-2}$ on 17 October and the $\Delta BLH\_MAX$ was more than others (more than 400 m). Owing to the significant impact, we took

the model result of 17 October as an example to study the relationship among surface ozone, BC, and BL development. From 10:00 to 14:00 on 17 October, the average distribution (Figure 3c) of BC concentrations was inhomogeneous due to the inhomogeneous distribution of BC emissions (Qin and Xie, 2012) and wind fields at ground level. The northern areas were controlled by southerly winds, whereas northeast winds blew over the southern areas. In the central areas (most parts of Jiangsu and Anhui), winds were weak, which is favourable for the accumulation of air pollutants. BC was therefore mainly

concentrated in Nanjing and the surrounding areas. Conversely, surface ozone showed low concentration over this region, suggesting that BC and ozone at ground level showed opposite distributions over Nanjing.

## 3.3 The BC-BL interaction and its effects on photolysis rate and ozone precursors

At high concentrations of BC in the atmosphere, the incident shortwave radiation is attenuated by BC. Consequently, other meteorological elements are affected. Figure 4a presents the vertical profiles of BC concentration (black solid line) and

changes in related meteorological elements (Exp_WF−Exp_WFexBC) induced by BC at 10:00, 12:00, and 14:00. The vertical profiles of the atmospheric attenuation of shortwave radiation (ΔSW; blue solid line), which is defined as the gradual vertical loss in intensity of the incident solar radiation induced by BC, shows a small reduction above 1.2 km where



BC concentrations are low. With BC increasing downwards, ΔSW becomes larger. A greater reduction in shortwave radiation could be observed at a height of 1 km than at the lower adjacent level. This suggests that, when BC concentrations are similar in adjacent layers, solar radiation absorption by BC is more efficient at higher altitudes than that at lower altitudes (Ferrero et al., 2014; Ding et al., 2016). It also should be noted that when BC concentration is sufficiently high, large

amounts of BC will absorb more shortwave radiation; hence, the attenuation of shortwave radiation displayed a second peak below the height of 600 m. Because of the absorption of solar radiation induced by BC, the heating rate of shortwave increase correspondingly. More shortwave absorption leads to a greater increase in heating rate (magenta solid line), which will lead to a more rapid rise in temperature above the BL than in the upper BL (Figure 4a 10:00 and 12:00). Under the influence of BC (in Exp_WF), the Equivalent Potential Temperature (EPT; grey solid line) exhibited a lower value at the top

of the BL (black dashed line) than that at higher altitudes which suggested the temperature inversion was formed above the BL. This will increase atmospheric stability and suppress the development of the BL (black dashed line) at 10:00 and 12:00. Conversely, the EPT in Exp_WFexBC (orange solid line) did not exhibit this property and the BL (red dashed line) could develop rapidly, reaching 1 km altitude by 12:00. At 14:00, the BL in Exp_WFexBC was almost fully developed, whereas the BL in Exp_WF was still developing and rising close to the BL in Exp_WFexBC.

As one of the most important ozone precursors, the photolysis of $NO_2$ directly contributes to the photochemical formation of ozone. Due to the attenuation of incident solar radiation induced by BC, the photolysis rate of $NO_2$ was reduced in the daytime ($\Delta J[NO_2]$; brown solid line in Figure 4b), and this is consistent with the results of previous studies (Dickerson et al., 1997; Li et al., 2005). $NO_2$ in the atmosphere is primarily formed through chemical production or directly emitted from surface sources (e.g. industry, transportation, and soil) and is usually confined in BL. As a result of BL suppression caused

by BC, more $NO_2$ is confined below the BL, as shown in Figure 4b ($\Delta NO_2$; green solid line). Similar to $NO_2$, other ozone precursors (e.g. NO, VOCs) also showed the same variations in vertical distribution (figures are not shown). The chemical production of tropospheric ozone is affected by both photolysis rate and the concentrations of precursors (Crutzen, 1973). When only considering the photolysis rate, the reduction of $J[NO_2]$ will weaken the photochemistry and reduce ozone concentration. However, the chemical production of ozone is not only related to the photolysis rate, but also closely related

to ozone precursors. Using the online model system, with the aerosol-BL feedback mechanism, the influence of ozone precursors' changes will be observed and the change in chemical production of ozone may be different from previous offline model studies. In addition, the changes in BL development could also affect the relevant physical processes in the BL, which will influence surface ozone. Quantitative changes in ozone contributions from chemical and physical processes will be discussed in the following section.

**3.4 Changes in surface ozone resulting from BC-BL interaction**

Because variations in ozone concentration are directly caused by physical and chemical processes, the mechanism of BC-BL interactions affecting ozone concentration can be determine through process analysis (Zhu et al., 2015). The following processes were considered in this work: (1) advection (ADV) caused by transport, which is highly related to winds and





ozone concentrations upwind; (2) vertical mixing (VMIX) caused by atmospheric turbulence and vertical ozone gradients, which is closely related to the development and variations of the BL (Zhang and Rao, 1999; Gao et al., 2017); and (3) chemistry (CHEM), which is the net value of chemical calculations including ozone chemical production and chemical reduction. The contribution of convection process, i.e. the contribution of ozone by dynamic and thermodynamic effects, was

5 negligible, and is not mentioned in this study. Complete details on the process analysis in WRF-Chem are available in Zhang et al. (2014), Gao et al. (2016), and the WRF-Chem user guide.

The simulated surface ozone concentrations in Exp_WF and Exp_WFexBC are shown in Figure 5a. As shown in the figure, surface ozone increased slowly from 08:00 to 09:00, and the concentrations were below 20 ppb. From 10:00, surface ozone started to increase rapidly, especially in Exp_WFexBC. Based on the processes contribution in Exp_WFexBC (Figure 5b),

VMIX contributed more than CHEM during the period from 10:00 to 14:00. Overall, the contribution of ADV was much less than those of VMIX and CHEM. The process contribution suggests that the increase in ozone is highly related to the contribution of ozone by vertical mixing. Because of the impacts of BC, the process contributions also changed. Differences in the contributions of the processes (Exp_WF−Exp_WFexBC) are presented in Figure 5c. From 10:00 to 12:00, changes in chemical contribution (CHEM_DIF) increased surface ozone at a rate of 3.1 ppb h$^{-1}$. However, VIMX_DIF decreased

surface ozone more significantly and at a greater rate of approximately -8.2 ppb h$^{-1}$. Changes in all the processes, as a result of the impacts of BC, further decreased surface ozone (Exp_WF) from 10:00 to 12:00 with the maximum reduction of 16.4 ppb at 12:00 (Figure 5a). In the afternoon (13:00 to 14:00), the CHEM_DIF was near to zero which shows little influence to the variation of surface ozone. However, the reduction of ozone became weaker and it was mainly because the effects of VMIX with BC is larger than those without BC (VMIX_DIF is positive), which offsets the reduction of surface ozone. After

14:00, there were no significant differences between the two ozone patterns.

Among all the changes in processes caused by BC, VMIX_DIF accounted for more than 50% of the net contribution variation (NET_DIF), indicating that changes in vertical mixing are the primary means of BC impacting surface ozone. In addition, although $J[NO_2]$ was reduced by the impacts of BC, changes in CHEM at the surface increased before noon. The reduction in photolysis rate should reduce the chemical production of ozone; however, changes in ozone precursors will also

influence the chemical production of ozone. Under the influence of BC on the atmosphere, more ozone precursors (i.e. $NO_2$) are confined below the BL due to the suppression of BL development (Figure 4b). With the photolysis of a large amount of $NO_2$ at the surface, ozone chemical production will be enhanced and may offset the influence of the reduction in photolysis rate. Thus, an enhanced CHEM_DIF was shown near the surface (3 ppb h$^{-1}$) from 10:00 to 12:00.

The development of the BL directly affects vertical mixing and the distribution of ozone in the BL (Zhang and Rao, 1999).

Here, we present VMIX distributions with (Figure 6a) and without (Figure 6b) the effects of BC to discuss the impacts of BC-BL interactions on surface ozone. The BL with (black line) and without (red line) the effects of BC are also presented in Figure 6. In general, ozone concentration is higher in upper layers before sunrise and in the early morning over polluted regions (Wang et al., 2015). When the BL develops, turbulence exchanges ozone in the vertical direction, resulting in entrainment of the abundant ozone aloft down to the surface (Figure 6a and b). Under the effects of BC, the slowly



developing BL weakens turbulence and mixing height, which leads to decreased downward entrainment of ozone (-8.2 ppb h⁻¹) from 10:00 to 12:00. Differences in VMIX (VMIX_DIF) between Exp_WF and Exp_WFexBC showed the changes in the contribution of ozone from vertical mixing caused by the effects of BC. The vertical distribution of VMIX_DIF is shown in Figure 6c.

Similar to VMIX_DIF, changes in the contribution of ozone from chemistry (CHEM_DIF) are presented in Figure 6d. As discussed above, the BC-BL interaction leads to reduced photolysis rates and more accumulation of ozone precursors under the BL. The photochemical production of ozone is related to both photolysis rate and ozone precursors. The reduction of photolysis rate will surely decrease photochemical rate. However, the photolysis of a large amount of $NO_2$ at the surface may offset the influence of the reduction in photolysis rate and enhance the chemical production of ozone. Thus, enhanced CHEM contribution could be shown near the surface (with positive CHEM_DIF of 3 ppb h⁻¹) from 10:00 to 12:00. In upper layers, less $NO_2$ is mixed up owing to the suppression of BL, leading to a decreased CHEM contribution (with negative CHEM_DIF between -9.4 and -2.1 ppb h⁻¹) at a height of 500–1000 m from 11:00 to 14:00. Under the effects of BC, the ADV decreases slightly within the BL before noon (Figure 6e). In Figure 6f, the difference of the net contribution of all processes (NET_DIF) represents the variation in ozone caused by the effects of BC. Comparing Figures 6c to 6f, the similar distributions and the significant ratio of VMIX_DIF to NET_DIF suggest that the change in vertical mixing induced by BC has the greatest influence on surface ozone.

Due to BC-BL interactions, VMIX in Exp_WF is weakened before noon but enhanced in the afternoon (Figure 7a). The contribution of VMIX depends on the vertical gradients of ozone and the turbulent exchange coefficient. Vertical gradients of ozone (solid lines in Figure 7b) showed similar profiles in both experiments during the early stage of BL development before noon. In the afternoon, the ozone gradient (red solid line in Figure 7c) decreased significantly without the effects of BC in the mature stage of BL development, whereas the ozone gradient remained large at the top of the BL under the effects of BC (black line in Figure 7c). The effects of BC can lead to a significantly smaller turbulence exchange coefficient (black line with square in Figure 7b), lowering the entrainment of ozone from higher altitudes to the surface, which indicates that the contribution of VMIX in Exp_WF is smaller before noon. In the afternoon, the faster developing BL without the effects of BC forms a uniformly mixed ozone profile, with very small vertical gradients. As a result, the contribution of VMIX in Exp_WFexBC is smaller, although the turbulence exchange coefficient is still large. In contrast, with the impacts of BC, larger ozone gradients at the top of the BL result in stronger entrainment of ozone down to the surface which leads to the ozone contribution of VMIX in Exp_WF became larger at surface.

## 4 Conclusions

In this study, measured data of BC and ozone in Nanjing, an important industrial and economic centre in the YRD region of China, showed that ozone concentration increases slowly from 10:00 to 14:00 when the BC concentration is relevant high. Because of the limited of measurement, the WRF-Chem model was applied to study the impacts of BC-BL interactions on




surface ozone in Nanjing. Acceptable agreement was achieved between observed and simulated results for both meteorological and chemical variables, suggesting that the WRF-Chem model has the capability to accurately reproduce meteorological factors and air pollutants in this study.

The model results show that, when high concentrations of BC are confined over Nanjing, the incident solar radiation is absorbed. The absorbed radiation heated the air above the BL and suppressed the development of the BL. Because of the BL suppression, the contribution of chemical and physical processes to ozone (photochemistry, vertical mixing and advection) will change correspondingly. In particular, BL suppression leads more ozone precursors (e.g. $NO_2$) being confined below the BL. Although the photolysis rate is reduced as a result of the solar-radiation absorption induced by BC, more ozone precursors enhanced the chemical production of surface ozone before noon. However, as a more significant impact, the

suppressed BL weakened the turbulence and entrained very little ozone aloft down to the surface. As a result, surface ozone decreased before noon, with the maximum reduction reaching 16.4 ppb at 12:00. In the afternoon, the changes in chemical production were small which influence inconspicuously to surface ozone. However, physical process still influences the surface ozone significantly. Due to the delay of the BL development, the vertical ozone structure showed larger vertical gradients near the top of the BL, and more ozone aloft could be entrained down to the surface to offset the surface ozone

reduction before noon.

Comparing all the changes in the contributions of processes induced by the effects of BC, the change in vertical mixing caused by BL suppression plays a more important role on surface ozone. Our results enhance the great impacts of BC-BL interactions on surface ozone. The aerosol-BL-gas relationship will provide new insights in ozone pollution control in China.

**Acknowledgements**

This work is supported by grants from the National Key Research and Development Program of China (2016YFA0602003), National Natural Science Foundation of China (91544229). All the model results in this study were calculated by the computational resources provided by Nanjing University of Information Science & Technology (NUIST). For the observations used in this paper, the data of temperature, wind direction, and wind speed can be accessed from the website http://data.cma.cn/. Hourly concentrations of $O_3$, $NO_2$, and $PM_{2.5}$ can be downloaded from

http://113.108.142.147:20035/emcpublish. The shortwave irradiance and BC concentrations can be accessed from the Comprehensive Meteorological Observation Base in Nanjing, which is attached to the CMA, after sending application and receiving the permission. All these observations and model outputs used in this study are available. Readers can access the data directly or by contacting B. Zhu via binzhu@nuist.edu.cn.



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



**Table 1: Major model configuration options used in the simulations**

| Item | Selection |
| --- | --- |
| Long wave radiation | RRTMG |
| Shortwave radiation | RRTMG |
| Microphysics scheme | Lin scheme |
| Boundary layer scheme | Yonsei University (YSU) scheme |
| Land surface option | Noah land-surface model |
| Photolysis scheme | Fast-J Photolysis |
| Dry deposition | Wesely scheme |





**Table 2: Statistical metrics for meteorological variables from 10 to 26 October 2015. The benchmarks follow the recommended values reported by Emery et al. (2001). The values that not meet the criteria are denoted in bold.**

| Variables | | Nanjing | Zhenjiang | Maanshan | Hefei | Wuxi | Criteria |
|---|---|---|---|---|---|---|---|
| WD (°) | IOA | 0.94 | 0.95 | 0.91 | 0.92 | 0.94 | |
| | MB | -0.54 | **-11.72** | -5.83 | **-15.29** | **-11.39** | $\leq \pm 10$ |
| | RMSE | 42.53 | 39.83 | 53.78 | 52.43 | 45.03 | |
| WS (m s$^{-1}$) | IOA | 0.72 | 0.76 | 0.66 | 0.65 | 0.63 | $\geq 0.6$ |
| | MB | 0.26 | -0.4 | 0.01 | **0.72** | **0.87** | $\leq \pm 0.5$ |
| | RMSE | 1.11 | 1.2 | 1.43 | 1.21 | 1.49 | $\leq 2$ |
| T2 (℃) | IOA | 0.93 | 0.87 | 0.95 | 0.91 | 0.92 | $\geq 0.8$ |
| | MB | **-1.71** | **-1.93** | -0.33 | **-2.03** | **-0.88** | $\leq \pm 0.5$ |
| | RMSE | 2.44 | 2.8 | 1.79 | 2.5 | 2.31 | |
| SW (×10$^3$ W m$^{-2}$) | IOA | 0.96 | | | | | |
| | MB | 0.01 | | | | | |
| | RMSE | 0.09 | | | | | |



**Table 3: Statistical metrics for air pollutants from 10 to 26 October 2015. Criteria for ozone and PM$_{2.5}$ are suggested by EPA (2005) and EPA (2007). The values that do not meet the criteria are denoted in bold.**

| Variables | | Nanjing | Zhenjiang | Maanshan | Hefei | Wuxi | Criteria |
|---|---|---|---|---|---|---|---|
| O$_3$ (ppb) | IOA | 0.91 | 0.85 | 0.84 | 0.93 | 0.84 | |
| | MNB | **-0.32** | 0.02 | -0.09 | 0.05 | **0.24** | ≤ ±0.15 |
| | MFB | -0.59 | -0.61 | -0.52 | -0.10 | -0.31 | |
| NO$_2$ (ppb) | IOA | 0.81 | 0.77 | 0.70 | 0.73 | 0.79 | |
| | MNB | -0.13 | 0.05 | 0.51 | -0.17 | -0.32 | |
| | MFB | -0.25 | -0.05 | 0.15 | -0.34 | -0.49 | |
| PM$_{2.5}$ (µg m$^{-3}$) | IOA | 0.62 | 0.76 | 0.77 | 0.59 | 0.67 | |
| | MNB | 0.12 | 0.13 | 0.10 | 0.11 | 0.10 | |
| | MFB | -0.04 | -0.01 | -0.01 | -0.02 | -0.02 | ≤ ±0.6 |
| BC (µg m$^{-3}$) | IOA | 0.67 | | | | | |
| | MNB | 0.76 | | | | | |
| | MFB | 0.38 | | | | | |





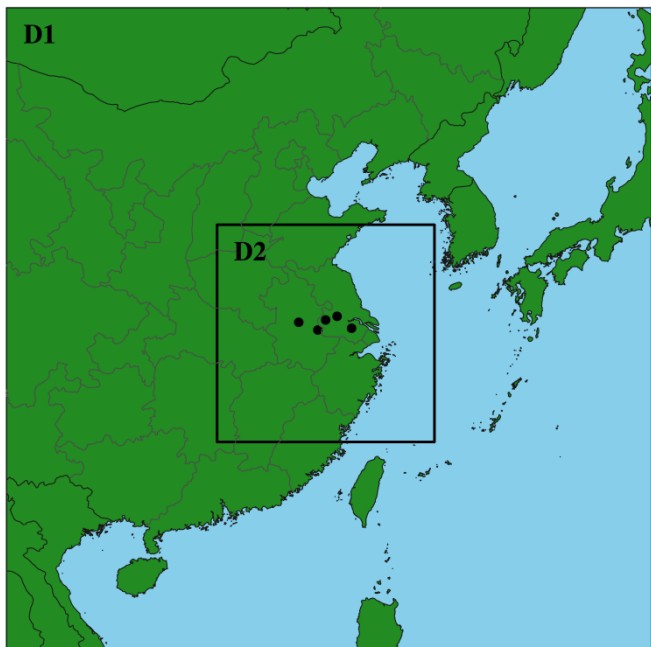

**Figure 1: Map of the two model domains. The locations of the observation sites used for model evaluation are presented as dots. From west to east, the sites are located in Hefei, Maanshan, Nanjing, Zhenjiang, and Wuxi.**



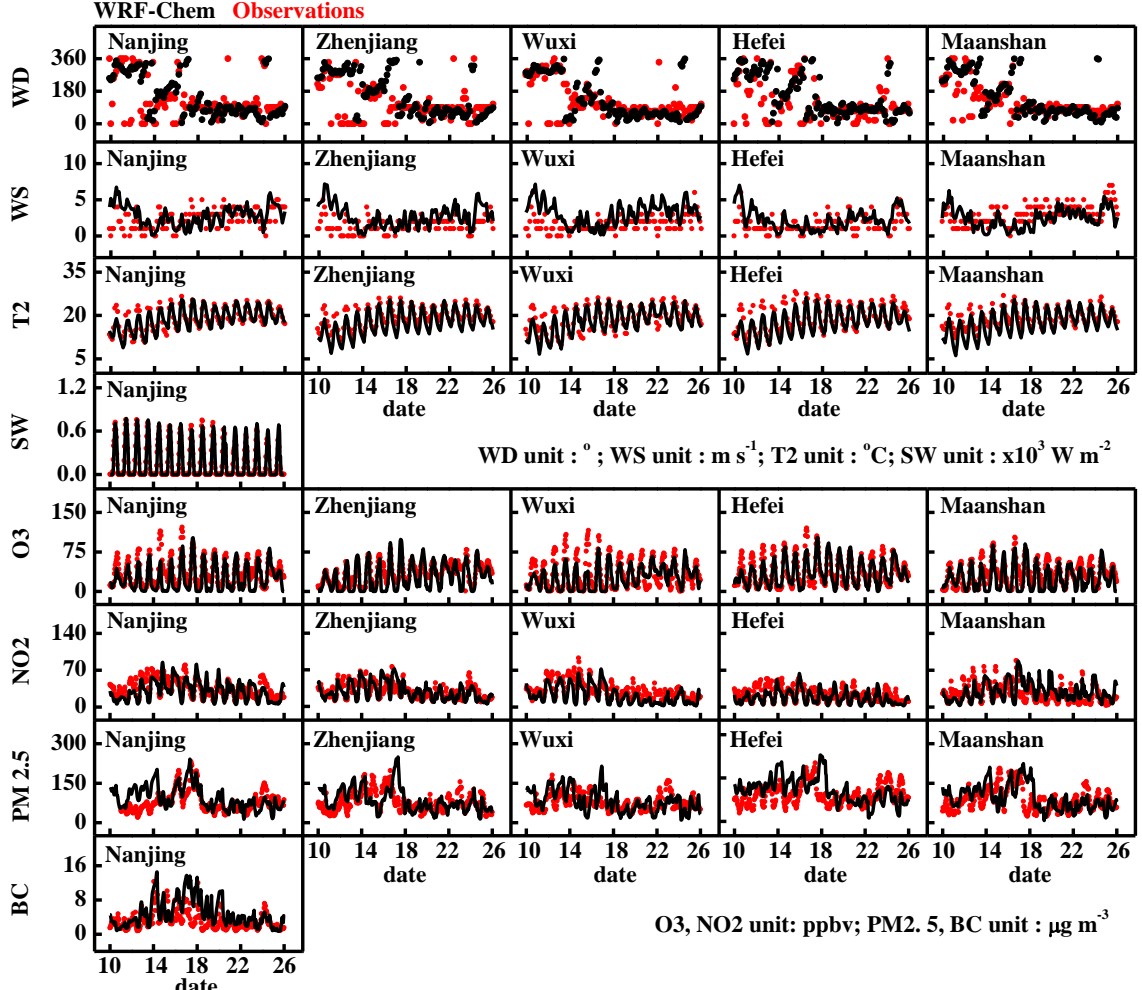

**Figure 2: Time series comparisons between observed and simulated variables from 10 to 26 October 2015. (WD = Wind Direction; WS = Wind Speed; T2 = Temperature at 2 m above the surface; SW = Shortwave; O3 = Ozone; NO2 = Nitrogen dioxide; PM2.5 = particulate matter with diameter less than or equal to 2.5 μm; BC = Black Carbon)**





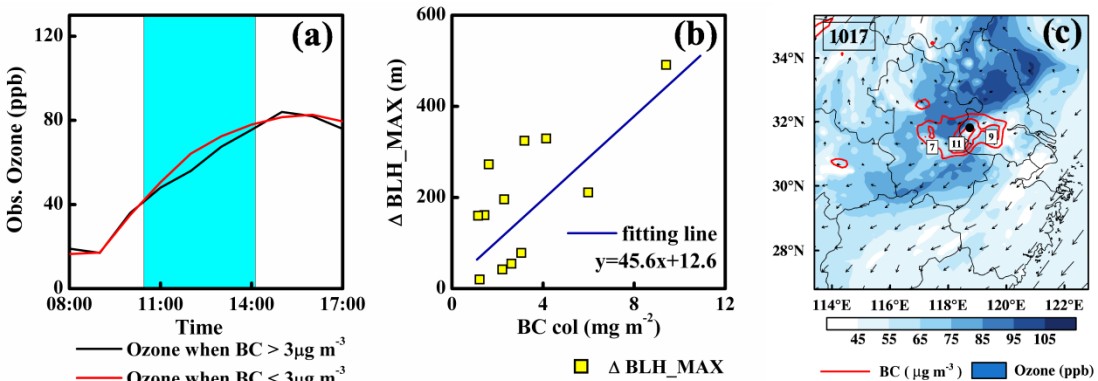

**Figure 3: (a) Observational averaged ozone concentrations with BC concentrations lower (red) and higher (black) than the monthly mean value during October 2015; (b) simulated maximum BLH changes (yellow square) induced by BC; (c) Average distribution of surface BC (red contour) and ozone concentrations (blue colour fill), showing wind fields from 10:00 to 14:00 local on 17 October 2015. Black dot denotes the location of the observation site.**





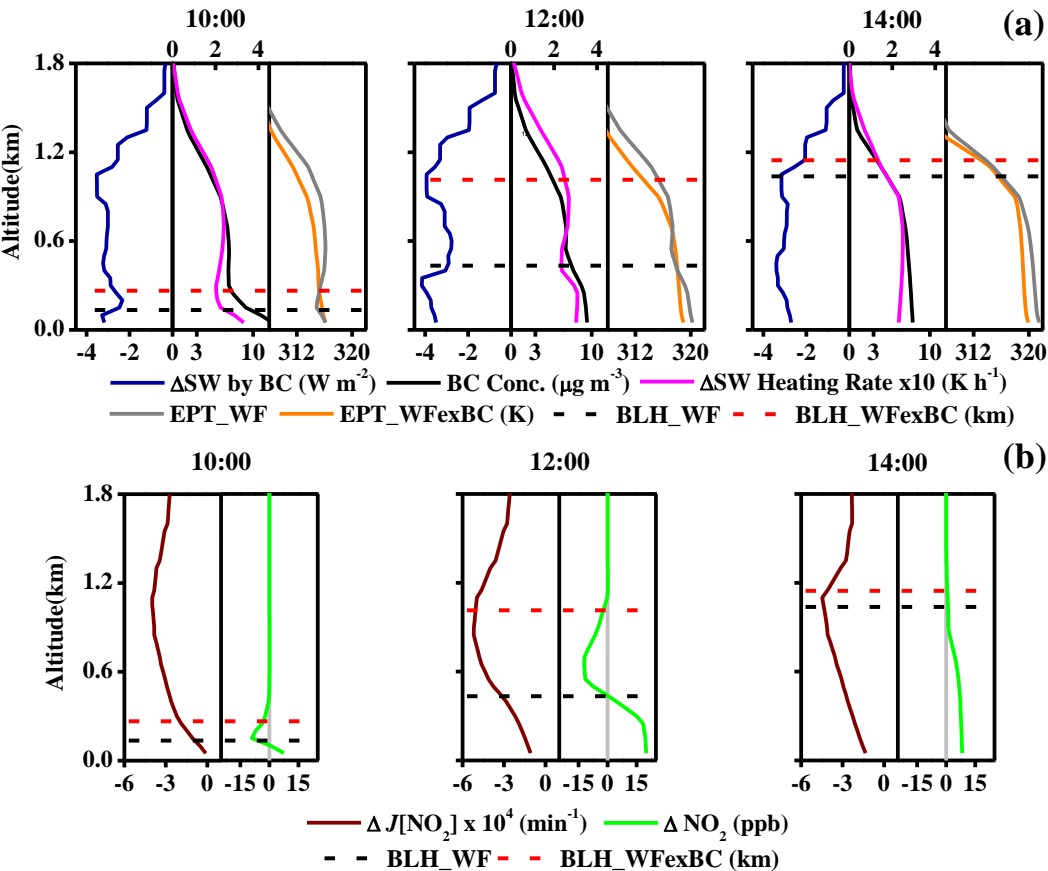

**Figure 4: Vertical distributions of (a) averaged BC concentrations (black line; unit: μg m⁻³), attenuation of incident shortwave radiation induced by BC (blue line; unit: W m⁻²), change of shortwave heating rate induced by BC (magenta line: unit: K h⁻¹), Equivalent Potential Temperature (EPT; unit: K) of the two experiments (dark grey line for Exp_WF and orange line for Exp_WFexBC); (b) average changes in *J*[NO₂] (brown line; unit: min⁻¹) and NO₂ (green line; unit: ppb) induced by BC in Nanjing at 10:00, 12:00 and 14:00. BLHs of the two experiments (black and red dashed line denotes Exp_WF and Exp_WFexBC) are presented in (a) and (b).**



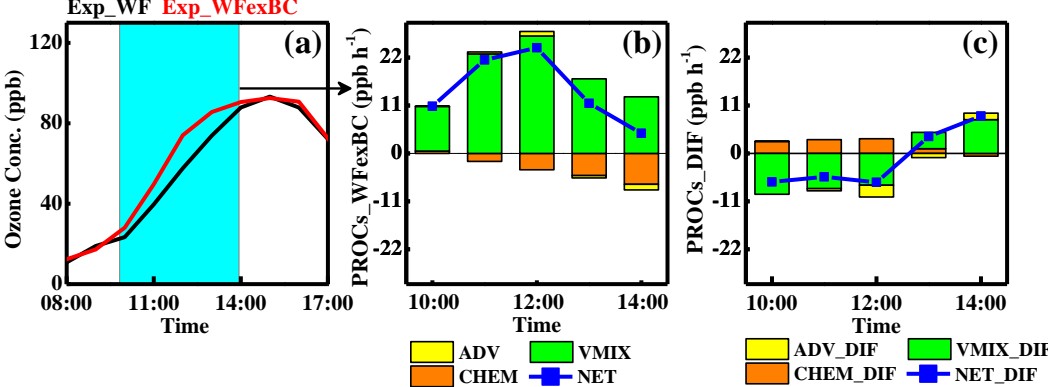

**Figure 5: (a)** Time series of simulated ozone concentration (unit: ppb) from 08:00 to 17:00; **(b)** hourly process contributions in the Exp_WFexBC experiment from 10:00 to 14:00; and **(c)** variations in process contributions caused by BC-BL interactions (Exp_WF−Exp_WFexBC) from 10:00 to 14:00. Cyan shades in (a) highlight the ozone differences induced by BC. In (b): ADV is advection, VMIX is vertical mixing, and CHEM is chemistry; the blue line with squares denotes the net value of the processes (NET); Variations in each process and the net contribution are denoted by ADV_DIF, VMIX_DIF, CHEM_DIF and NET_DIF. Units in (b) and (c) are ppb h$^{-1}$.

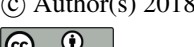



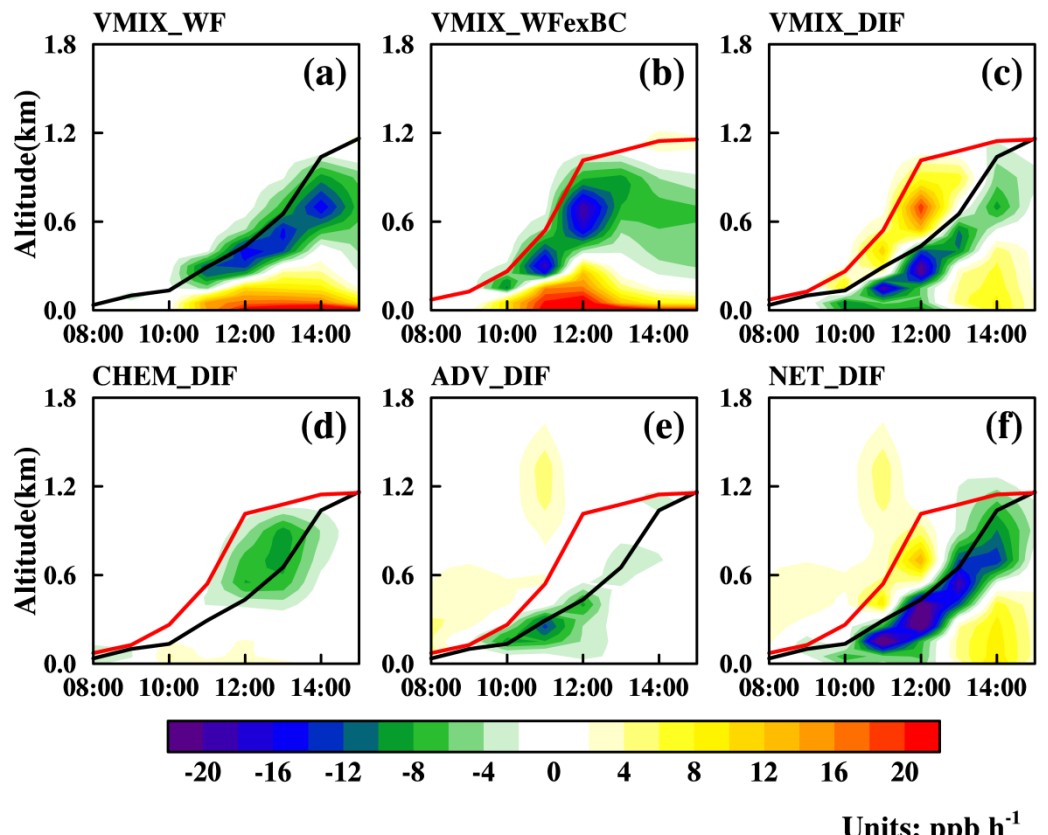

**Figure 6: Vertical distribution of ozone contribution from the vertical mixing process in Exp_WF (a) and Exp_WFexBC (b). Changes of VMIX (VMIX_DIF; c), CHEM (CHEM_DIF; d), and ADV (ADV_DIF; e) between the two experiments, and the net value of the changes from all processes (NET_DIF; f) from 08:00 to 15:00. Units are ppb h⁻¹. The black and red lines denote the BLH in Exp_WF and Exp_WFexBC.**





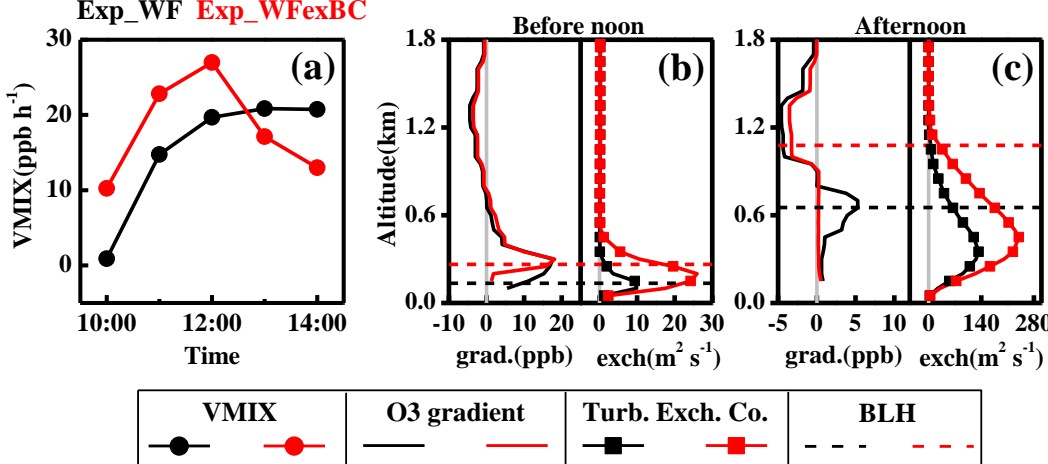

**Figure 7: (a) Time series of vertical mixing contribution (line with circle; unit: ppb h$^{-1}$) from 10:00 to 14:00 and profiles of ozone vertical gradients (solid line; unit: ppb) and turbulence exchange coefficient (line with square; unit: m$^2$ s$^{-1}$) in Exp_WF (black) and Exp_WFexBC (red) before noon (b) and afternoon (c). BL heights (BLH; dashed line; unit: km) of Exp_WF and Exp_WFexBC are denoted in (b) and (c)**