# Peer review of "Effects of black carbon and boundary layer interaction on surface ozone in Nanjing, China"

_Atmospheric Chemistry and Physics, 2017_

## Referee Comment (RC1) · Anonymous Referee #2 · 29 Jan 2018

Review of "Effects of black carbon and boundary layer interaction on surface ozone in Nanjing, China" by Gao et al.

The main focus of the paper is to discuss the effect of BC on solar radiation, PBL, and surface ozone. Some interesting results are found by the authors. For example, they find that the absorbing effect of BC heats the air above the BL and suppresses PBL development, which eventually leads to changes in the contributions of ozone through chemical and physical processes (photochemistry, vertical mixing, and advection). The heavy aerosol pollution is a very important issue in China, and this paper fits well in the scope of ACP. However, there are some comments regarding the paper, and the authors should carefully address these comments before the publication.

(1) Fig. 2 is too crowded. In order to better evaluate the model simulation in details,

the Authors should enlarge the figure. I suggest it can be separated to 2 figures. One figure only contains meteorological parameters (Fig. 2a) and another is for chemical species (Fig. 2b). (2) Fig. 3 has a similar problem. The Fig. 3C is impossible to read. It should be an individual panel. (3) Why there is a consistent heating by BC around 1.2 km, especially at 10am. If it is due to residual layer of BC, the authors should explain it in more details. (4) In the introduction, the Authors should reference the work by Tie et al. (2005). Although it used a global model, it is an early work to discuss the effect of aerosols (including BC) on photochemistry and ozone. Also in Tie et al. (2017), they found that the moister plays important roles on PBL development, especially in the aged aerosols, including BC. The Authors should state this point in the Introduction. (5) In previous works (Tie et al., 2009), in large cities in eastern China, NOx concentrations are very high. As a result, increase in NOx concentrations lead to decrease in ozone concentrations in the center of cities. However, in rural areas, the concentrations of NOx decrease rapidly, and increase in NOx concentrations lead to increase in ozone concentrations of ozone. In the analysis of paper, the Authors should discuss this point in more details.

References

Tie, X., S. Madronich, S. Walters, D.P. Edwards, P. Ginoux, N. Mahowald, R.Y. Zhang, C. Lou, and G. Brasseur, Assessment of the global impact of aerosols on tropospheric oxidants, J. Geophys. Res., 110 (D03204), doi:10.1029/2004JD005359, 2005.

Tie, X., R.J. Huang, J.J. Cao, Q. Zhang, Y.F. Cheng, H. Su, D. Chang, U. Pöschl, T. Hoffmann, U. Dusek, G. H. Li, D. R. Worsnop, C. D. O'Dowd, Severe Pollution in China Amplified by Atmospheric Moisture, Sci. Rep. 7: 15760 | DOI:10.1038/s41598-017-15909-1, 2017.

Tie, X., FH. Geng. L. Peng, W. Gao, and CS. Zhao, Measurement and modeling of O3 variability in Shanghai, China; Application of the WRF-Chem model, Atmos. Environ., 43, 4289-4302, 2009.

---

## Referee Comment (RC2) · Anonymous Referee #1 · 13 Mar 2018

This manuscript presents some very interesting results on the effect of black carbon on boundary layer development

and surface ozone. Most previous studies show the effects of black carbon on haze pollution or ozone photochemical

production, this study will provide more insights on ozone formation in China given the high balck carbon loadings.

The figures are of high quality while the presention quality needs improvements. I would ecourage the authors to find

native speaker or people with good English to polish the language. For the presented materials, I only have minor comments.

right

[Figure]

Specific comments: 1 the address for 5th institution should be 'Tropical and Marine Meteorology' 2 The conclusions in abstract are a little messy, please reorganzie your findings. 3 Page 9 line 17: Please define how you calculate the ozone gradient. 4 Page 10: the less ozone near the surface is very likely caused by less ozone production aloft (Figure 6d), not weakened tubulence. Please provide a more comprehensive exlanation for this. 5 Please be careful when interpret the surface VMIX term, because it also includes information of chemical production above surface. For example, if chemical production above surface larger and higher ozone above surface, it will lead to a positive VMIX term at the surface.

---

## Author Comment (AC1) · 14 Apr 2018

According to the comments of referee #1, we have carefully replied them one by one. Please check the zip file named referee #1. The response to the comments can be found in the file named reply to referee #1. The revised manuscript is also included in the zip file.

Please also note the supplement to this comment:
https://www.atmos-chem-phys-discuss.net/acp-2017-1177/acp-2017-1177-AC1-supplement.zip

2018.

---

## Author Comment (AC2) · 14 Apr 2018

according to the comments of referee #2, we have carefully replied them one by one. Please check the zip file. The response to the referee #2 is in the file named reply to referee #2. the revised manuscript and the supplement are also included in the zip file.

Please also note the supplement to this comment: https://www.atmos-chem-phys-discuss.net/acp-2017-1177/acp-2017-1177-AC2-supplement.zip

---

## Author Response (AR1)

Dear editor and referees,

We thank you for your time on our paper. The comments and suggestions are useful to improve the quality of the manuscript. Herein we present the replies to all the comments on our manuscript named "Effects of black carbon and boundary layer interaction on surface ozone in Nanjing, China"

Referee#1

(1) The address for 5th institution should be 'Tropical and Marine Meteorology'.

Reply: Follow the referee's comment. We have rechecked the address of the 5th institution and corrected it in the revised manuscript.

(2) The conclusions in abstract are a little messy, please reorganize your findings.

Reply: Thank you for your comment. Our main conclusion of our study is that, with the impacts of BC, the surface ozone reduced before noon which is primarily caused by the changes in the ozone contribution from chemical and physical processes. Among the changes in these processes, the change in vertical mixing process takes major responsibility for the reduction of surface ozone. We have adjusted the conclusions in abstract. And we believe that the conclusion will be more clear and easy to following. Please check the new abstract in the revised manuscript.

(3) Page 9 line 17: Please define how you calculate the ozone gradient.

Reply: Follow this comment. The vertical gradient of ozone is the difference of the ozone concentration between every two adjacent vertical layers. Because of the sigma coordinate in vertical direction in WRF-Chem, the ozone concentration is firstly interpolated in vertical direction with height interval of 50m. And then, the vertical gradient of ozone concentration is calculated as the Equation:

$$vertical\ gradient = O_3(h_{i+1}) - O_3(h_i)\ \ (i = 1, 2, 3 \ldots\ldots)$$

where $h_i$ and $h_{i+1}$ are the heights of layer $i$ and $(i + 1)$. The $h_{i+1}$ equals to $h_i + 50$. $O_3(h_i)$ is the ozone concentration at the height of $h_i$. The definition of ozone gradient and the equation have been added

into the revised manuscript. Please check them in page 9 (lines from 31 to 34) and page10 (lines from 1 to 3).

(4) Page 10: the less ozone near the surface is very likely caused by less ozone production aloft (Figure 6d), not weakened turbulence. Please provide a more comprehensive explanation for this.

Reply: Thank you for your comment. The less ozone chemical production aloft will influence the surface ozone. For example, the reduction of chemical contribution will change the ozone vertical gradients which could influence the vertical mixing of ozone which occurs in the BL and finally reduce the surface ozone. However, according to our results, the CHEM_DIF aloft occurred one hour later than the reduction of surface ozone did. In addition, the value of CHEM_DIF was small whereas the value of VMIX_DIF in the BL was large (-18 - -8 ppb h$^{-1}$) which suggested that the less chemical production aloft impacted limitedly on vertical mixing and surface ozone. Besides ozone gradients, VMIX is closely related to the turbulence. As Figure 7b shows, the turbulence exchange coefficient with the impacts of BC was much smaller than that without the impacts of BC. It suggested that, when ozone gradients in the two cases were similar with each other in the morning, the much weaker turbulence would entrain much less ozone down to surface and made the ozone reduction. Thus, it could be concluded that the reduction of surface ozone was primarily caused by the weakened turbulence. We have added more detailed discussion of this section and please check the details in pages from 8 to10 in the revised manuscript.

(5) Please be careful when interpret the surface VMIX term, because it also includes information of chemical production above surface. For example, if chemical production above surface larger and higher ozone above surface, it will lead to a positive VMIX term at the surface.

Reply: Thanks, we agree with your comment. The contribution from VMIX at surface is related to the turbulence and ozone vertical gradient above surface. Chemical production above surface will influence ozone vertical gradient. Thus, ozone exchanging in vertical direction, the VMIX at surface is surly includes the information of chemical production above surface. In our study, we primarily talked about the changes in processes contributions caused by BC. In figure 6d, with the impacts of BC, chemical

contribution aloft decreased (with decreased rate between -9.4 and -2.1 ppb h[-1]) from 11:00 to 14:00. This reduction would change the vertical gradients of ozone and further influence the vertical mixing of ozone. However, the significant change in vertical mixing in the BL (-18 to -8 pph h[-1]) suggested that the changes in chemical process cause limited influence on the changes in vertical mixing process. According to our discussion, the weakened turbulence and the suppressed BL, which being caused by the impacts of BC, were more important to the change of VMIX.

Referee#2

(1) Fig. 2 is too crowded. In order to better evaluate the model simulation in details, the authors should enlarge the figure. I suggest it can be separated to 2 figures. One figure only contains meteorological parameters (Fig. 2a) and another is for chemical species (Fig. 2b).

Reply: Follow the referee's comment. The figure 2 is separated to 2 parts, Figure 2a only contains the meteorological parameters and Figure 2b only contains the chemical species. Please check the new Figure 2 in the revised manuscript.

(2) Fig. 3 has a similar problem. The Fig. 3C is impossible to read. It should be an individual panel.

Reply: Follow the referee's comment. Figure 3 is rearranged, Fig. 3a and 3b are set as a panel and Fig. 3c is set as an individual panel. Please check the new Figure 3 in the revised manuscript.

(3) Why there is a consistent heating by BC around 1.2 km, especially at 10am. If it is due to residual layer of BC, the authors should explain it in more details.

Reply: Thank you for your comment. The consistent heating above BL is exactly due to the impacts of BC in the residual layer. We have added the explanation in the revised manuscript and please check the details in page 7 lines from 15 to 22.

(4) In the introduction, the authors should reference the work by Tie et al. (2005). Although it used a global model, it is an early work to discuss the effect of aerosols (including BC) on photochemistry and

ozone. Also in Tie et al. (2017), they found that the moister plays important roles on PBL development, especially in the aged aerosol, including BC. The authors should state this point in the introduction.

Reply: Thanks. These references are very helpful to our study. They have been cited in the introduction. Please check the details in the revised manuscript.

(5) In previous works (Tie et al., 2009), in large cities in eastern China, $NO_x$ concentrations are very high. As a result, increase in $NO_x$ concentrations lead to decrease in ozone concentration of in the center of cities. However, in rural areas, the concentrations of $NO_x$ decrease rapidly, and increase in $NO_x$ concentrations lead to increase in ozone concentrations of ozone. In the analysis of paper, the authors should discuss this point in more details.

Reply: Thank you for your comment. The photochemical production of ozone is not only related to the photolysis rate, but also related to the concentrations of ozone precursors ($NO_x$ and VOCs). Since the suppression of BL, the concentrations of $NO_x$ and VOCs increased at surface (figure S2). It should be noted that, the ratios of VOCs/$NO_x$ also increased which suggested that VOCs increased more

15 significantly than NOx did. In addition, the little changes of the ratio of HCHO/$NO_y$ indicated that ozone still formed under the VOC-limited conditions. In this case, the increase of VOCs was favorable for ozone chemical formation. In our study, ozone contributions from chemistry were enhanced with the impacts of BC from 10:00 to 12:00. During this period, the photolysis rate was reduced which showed the reduce effect to the chemical production. Thus, the enhancement of ozone chemical production was

20 more likely related to the increase of ozone precursors. In order to interpreting this problem, we have added a figure (Figure S2) in the supplement and the relevant discussion has been added into the revised manuscript. Please check the relevant information in page 7 (lines from 24 to 28) page 8 (lines from 31 to 34) and page 9 (lines from 1 to 2).

[revised manuscript text omitted]